# Detailed Kinematic Analysis Reveals Subtleties of Recovery from Contusion Injury in the Rat Model with DREADDs Afferent Neuromodulation

**DOI:** 10.3390/bioengineering12101080

**Published:** 2025-10-04

**Authors:** Gavin Thomas Koma, Kathleen M. Keefe, George Moukarzel, Hannah Sobotka-Briner, Bradley C. Rauscher, Julia Capaldi, Jie Chen, Thomas J. Campion, Jacquelynn Rajavong, Kaitlyn Rauscher, Benjamin D. Robertson, George M. Smith, Andrew J. Spence

**Affiliations:** 1Department of Bioengineering, Temple University, Philadelphia, PA 19122, USA; tuh32834@temple.edu (K.R.); aspence@temple.edu (A.J.S.); 2Department of Biomedical Education and Data Science, Lewis Katz School of Medicine, Temple University, Philadelphia, PA 19140, USA; kathymkeefe@temple.edu; 3Merck & Co., Inc., North Wales, PA 19446, USA; george.moukarzel@merck.com (G.M.); sobrinerhd@gmail.com (H.S.-B.); juliacapaldi9@gmail.com (J.C.); 4Department of Biomedical Engineering, Boston University, Boston, MA 02215, USA; bcraus@bu.edu; 5Department of Neural Sciences, Lewis Katz School of Medicine, Temple University, Philadelphia, PA 19140, USA; jie.chen0121@temple.edu (J.C.); jacquelynn.rajavong@temple.edu (J.R.); george.smith@temple.edu (G.M.S.); 6Department of Pathology and Laboratory Medicine, Children’s Hospital of Philadelphia, Philadelphia, PA 19104, USA; thomas.campion@pennmedicine.upenn.edu; 7XCMR Inc., Narberth, PA 19072, USA; ben.robertson6786@gmail.com

**Keywords:** DREADDs or chemogenetics, designer receptors exclusively activated by designer drugs, clozapine-N-oxide (CNO), functional recovery after SCI, kinematics, plasticity, spinal cord injury

## Abstract

Spinal cord injury (SCI) often results in long-term locomotor impairments, and strategies to enhance functional recovery remain limited. While epidural electrical stimulation (EES) has shown clinical promise, our understanding of the mechanisms by which it improves function remains incomplete. Here, we use genetic tools in an animal model to perform neuromodulation and treadmill rehabilitation in a manner similar to EES, but with the benefit of the genetic tools and animal model allowing for targeted manipulation, precise quantification of the cells and circuits that were manipulated, and the gathering of extensive kinematic data. We used a viral construct that selectively transduces large diameter afferent fibers (LDAFs) with a designer receptor exclusively activated by a designer drug (hM3Dq DREADD; a chemogenetic construct) to increase the excitability of large fibers specifically, in the rat contusion SCI model. As changes in locomotion with afferent stimulation can be subtle, we carried out a detailed characterization of the kinematics of locomotor recovery over time. Adult Long-Evans rats received contusion injuries and direct intraganglionic injections containing AAV2-hSyn-hM3Dq-mCherry, a viral vector that has been shown to preferentially transduce LDAFs, or a control with tracer only (AAV2-hSyn-mCherry). These neurons then had their activity increased by application of the designer drug Clozapine-N-oxide (CNO), inducing tonic excitation during treadmill training in the recovery phase. Kinematic data were collected during treadmill locomotion across a range of speeds over nine weeks post-injury. Data were analyzed using a mixed effects model chosen from amongst several models using information criteria. That model included fixed effects for treatment (DREADDs vs. control injection), time (weeks post injury), and speed, with random intercepts for rat and time point nested within rat. Significant effects of treatment and treatment interactions were found in many parameters, with a sometimes complicated dependence on speed. Generally, DREADDs activation resulted in shorter stance duration, but less reduction in swing duration with speed, yielding lower duty factors. Interestingly, our finding of shorter stance durations with DREADDs activation mimics a past study in the hemi-section injury model, but other changes, including the variability of anterior superior iliac spine (ASIS) height, showed an opposite trend. These may reflect differences in injury severity and laterality (i.e., in the hemi-section injury the contralateral limb is expected to be largely functional). Furthermore, as with that study, withdrawal of DREADDs activation in week seven did not cause significant changes in kinematics, suggesting that activation may have dwindling effects at this later stage. This study highlights the utility of high-resolution kinematics for detecting subtle changes during recovery, and will enable the refinement of neuromechanical models that predict how locomotion changes with afferent neuromodulation, injury, and recovery, suggesting new directions for treatment of SCI.

## 1. Introduction

Spinal cord injuries (SCIs) typically cause severe reduction in quality of life, with no current highly effective treatment. SCIs often result from force trauma, but can also arise from nontraumatic origins like tumors, infections, or vascular problems. Damage to sensorimotor neural circuitry as a result of SCI causes a loss of function, with a distinct etiology in the acute and chronic phases. Various therapies to improve recovery from SCIs have been developed, largely targeted at strengthening and rewiring damaged and severed neural connections post-injury [1,2,3,4,5,6].

Approaches to improve recovery post-SCI include activity-based rehabilitation strategies. Treadmill training with and without body-weight support (BWSTT) is one such method [7]. These paradigms provide repetitive afferent input that is thought to engage spinal locomotor circuitry and encourage both functional recovery as well as plasticity. Another approach involves direct stimulation of muscle or nerves, functional electoral stimulation (FES) [8]. Research has consistently shown that treadmill training improves locomotor outcomes, with animals showing significant gains in Basso, Beattie, and Bresnahan (BBB) scores [9,10]. Treadmill-based rehabilitation has also been associated with increased axonal sprouting, reduced cavitation at the injury site, preservation of spinal tissue, as well as improvements in motor functions [10,11]. Importantly, variations in training paradigms influence cellular responses: some approaches appear to more strongly promote plasticity while others reduce astrocytic reactivity [10,11]. While treadmill training leverages repeated afferent input to engage spinal circuits, a complementary strategy is to directly modulate the sensory pathways themselves.

Falling into this category is a method that has recently come to the forefront with promising results: epidural electrical stimulation (EES). EES has demonstrated significant improvements in standing and locomotor capabilities in human patients post-SCI [6,12,13]. Studies have shown that EES targeting the dorsal roots of the lumbosacral segments promotes motor recovery and overall locomotion in animal studies and human SCI cases (human: [14,15,16,17,18]), (animal: [6,16,19]). EES is thought to work by stimulating primary afferents whose spikes impinge upon motor circuitry in the spinal cord, with the twofold beneficial results of (a) bringing motor outputs up to a more functionally useful level that partially compensates for reduced descending command and (b) inducing helping plasticity in motor circuitry. The latter plasticity is presently of great interest in the SCI community, where there is a need to know what new, modified, or strengthened connections are beneficial [6].

One approach to uncovering these mechanisms is to emulate the effects of EES in an animal injury model where detailed kinematic data and histological analyses can be carried out. To further enhance the scientific cutting power, genetically encoded actuators can be used to reversibly stimulate afferents that can be targeted to specific cell populations and labeled with tracers. Thus, we sought to mimic EES and selectively increase the excitability of large-diameter afferents within the lumbar dorsal root ganglia using one of the chemogenetic tools, the excitatory DREADD hM3Dq. DREADDs are engineering receptors that allow for temporary manipulation of the excitability of neurons [20]. DREADDs are activated by the binding of a specific, often engineered, ligand. The hM3Dq DREADD is activated by clozapine-*N*-oxide (CNO) [20]. The mechanisms by which the different DREADDs operate is under study, but many are thought to work via G-protein coupled pathways [20]. Here, expression of the hM3Dq DREADDs in the desired neuronal population (large diameter DRG afferents) was achieved through adeno-associated virus (AAV) delivery, and the inclusion of a fluorescent protein enables the identification of the targeted neurons [21]. In this and prior work we have validated this afferent transduction method (approximately ~42% of DRG neurons), an effect on a ladder behavioral task with activation [21], and the activation of second order neurons in the spinal cord with CFOS, a genetic marker of neuronal activity [20]. We used a dose of 2mg/kg of CNO based on the success of past studies using similar doses [21,22,23], and that is in the middle of the range published in the literature (~1 mg/kg to 5 mg/kg) [20]. DREADDs activation after intraperitoneal (IP) injection of CNO reaches a plateau after about 30 min and persists for approximately 2 to 3 h before declining [24].

Building on the observations of our prior study [21], the primary aims of this study were to examine in more detail the locomotor kinematics of recovery, and to do so in a more clinically relevant spinal cord injury model, the contusion injury. We utilized afferent excitation with the genetically encoded, excitatory activator DREADD hM3Dq, bilaterally transducing rat lumbar dorsal root ganglia (DRGs L3–L5) with direct intraganglionic injection. We used bilateral transduction in this study as opposed to the unilateral transduction of the prior study because the contusion injury is bilateral in nature. We applied a nine-week treadmill training with DREADDs afferent activation paradigm post-injury (Figure 1). We chose L3–L5 level DRGs because these segments provide broad sensory innervation to the hindlimbs [25]. We used an adeno-associated virus serotype 2 (AAV2) which has been previously shown to preferentially transduce large-diameter afferents when injected directly into the DRG [26]. In earlier work we confirmed its preferential transduction of large afferents and have also shown that this construct does not target thermal nociceptors as confirmed by Hargreaves Assay [21,23]. For construct design, we used the human synapsin promotor to restrict to neuronal expression and included an mCherry reporter for subsequent immunohistological verification.

We utilized a similar paradigm, but with a hemi-section injury model, in our prior work [21], and thus a secondary goal of this work was to compare the findings of these two studies that employed different injury models. The contusion injury is thought to be a more realistic model of injuries in humans [27], with more complicated etiology and sequelae following injury, and is bilateral in nature. The hemi-section model is typically used for more basic scientific questions because it gives less variability across animals, making it possible to resolve smaller effects, and because lateral hemi-sections allow for tracing of the routes of spinal cord pathways through the cord. In contrast, the contusion injury tends to involve more widespread tissue damage and secondary injury processes, which can complicate interpretation but also better mimics the heterogeneous nature of human spinal cord trauma [27]. We present the changes in stepping parameters and more detailed 3D kinematic parameters both during and at the conclusion of rehabilitation, where we find that DREADDs activation causes subtle but overall similar changes in the contusion model as opposed to the hemi-section model, but with some differences that may reflect the laterality and severity of the injury models.
Figure 1Apparatus, experimental design, and surgical approach. (**A**) Depiction of the treadmill kinematic recording apparatus and an example still frame. Five hindlimb markers were tracked with videos taken from two Ximea USB3 cameras at 250 frames per second (FPS) during flat treadmill locomotion at five separate speeds (16 cm/s, 20 cm/s, 24 cm/s, 28 cm/s, 32 cm/s). Marker tracking in 2D was conducted using DeepLabCut for each camera and then reconstructed to 3D with a custom Python-based pipeline [28]. (**B**) Tracked joints were the following five anatomical points on the hindlimb: anterior superior iliac spine (ASIS), Hip, Knee, Ankle, and metatarsal-phalangeal joint (MTP). (**C**) Timeline of experimental design spanning from baseline to week nine. Indicators show when animals received treadmill training, when kinematic recordings took place, surgery dates, and the ABAB withdrawal design, shown as ‘No CNO’ in weeks seven and nine. (**D**) Surgical methods overview—the spinal cord spanning segments T10 to L5 is shown with injection sites (needles) to represent DREADDs injection into DRGs while the red star indicates location of contusion injury. Shaded grey area denotes that segments T11–L2 were intact.
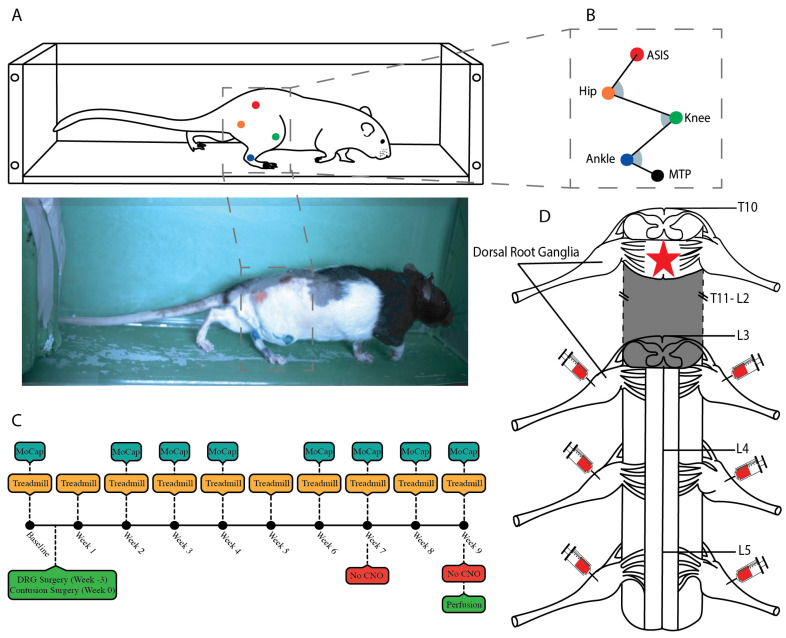


## 2. Materials and Methods (Figure 1)

### 2.1. Experimental Design

Animals were randomly assigned to either experimental or control groups, and a person outside the laboratory maintained a spreadsheet that connected the animal identification number to the treatment group, so that experimenters remained blinded during the study. Animals were acclimated to treadmill locomotion over approximately three sessions and then baseline kinematic data were collected (Figure 1A). Bilateral L3, L4, and L5 dorsal root ganglia (DRG) neurons were then transduced with excitatory DREADDs using direct intraganglionic injection in a recovery surgery (Figure 1D). A minimum of three weeks later, animals received a moderate severity 200 kDyn midline contusion injury at the T10 level. For the next nine weeks post injury, animals received treadmill training three times a week whilst DREADDs were activated (Figure 1C). In weeks one and two post-injury, animals were placed in the treadmill but typically could not locomote well until week three, although we were able to gather kinematic data on week two. We attempted treadmill training starting in week one, the week immediately post-surgery, but typically this was only briefly conducted due to the animal’s condition. We did not gather kinematic data in week one due to the severity of injury and lack of upright posture. Kinematic data were gathered on the treadmill in a separate session on weeks 2, 3, 4, 6, 7, 8, and 9; DREADDs activation was applied in all weeks except 7 and 9. To assess whether the presence of CNO is necessary for the locomotor effects associated with DREADD activation, treadmill locomotion was recorded with withdrawal of CNO at week seven and week nine post-injury.

### 2.2. Subjects

Female Long-Evans rats (225–250 g) were obtained from Charles River Laboratories Inc. (Wilmington, MA, USA) and housed in pairs with continuous access to food and water. The housing environment was maintained on a 12 h light/dark cycle, and all experimental procedures were performed during the light phase. All animal care and experimental protocols adhered to guidelines established by the Institutional Animal Care and Use Committee (IACUC; Protocol #5003, Andrew J. Spence) at Temple University and all guidelines established by the National Institutes of Health (NIH). Animals were monitored closely and euthanized if they met predetermined humane endpoints.

### 2.3. Surgical Procedures

All surgical procedures were conducted under aseptic conditions. Anesthesia was induced via intraperitoneal injection using a mixture of ketamine (100 mg/mL; Zetamine, Vet One, Boise, ID, USA), xylazine (100 mg/mL; AnaSed, Lloyd Laboratories, Shenandoah, IA, USA), and sterile saline. Anesthetic depth was maintained throughout the procedure with supplemental doses as needed. Muscle and skin layers were closed using 4-0 chromic gut sutures (DemeTECH, Miami Lakes, FL, USA) and surgical skin staples, respectively. Following surgery, animals received 10 cc of sterile saline, an antibiotic solution (0.5 g cefazolin powder reconstituted in sterile saline; Cat. No. NDC #0143-9923-90, Hikma Pharmaceutical USA, Inc., Eatontown, NJ, USA), and an analgesic (Rimadyl, 1 mg tablet; Cat. No. MD150-2, Bio-Serv, Flemington, NJ, USA).

### 2.4. DRG Injection Surgeries

Two categories of animals were utilized in this study: animals expressing excitatory DREADDs and administered CNO (referred to as “experimental”) and control animals that received CNO but were injected with a sham-mCherry constructed (referred to as “control”). In order to transduce large diameter afferent fibers, we directly injected AAV2-hSyn-hM3Dq-mCherry into lumbar dorsal root ganglia (DRG). Bilateral injections into the L3, L4, and L5 DRG were carried out. These nerve roots carry fibers whose afferents innervate hind-limb muscles, including those typically functioning as hip flexors and knee extensors (L3), hip adduction, knee extension/flexion, and further hip extension (L4), and knee flexion, hip extension, and ankle/toe movement (L5; [18,29,30]). We transduced the afferents using adeno-associated virus serotype 2 (AAV2) to preferentially target the larger diameter fibers [21,22,31]. To permit immunohistological analysis, viral constructs were engineered to include a fluorescent reporter (mCherry) under control of the human synapsin (hSyn) promoter. A midline dorsal skin incision approximately 5 cm long was made, beginning at the L1 spinal segment. The overlying fascia was carefully cut, and paraspinal muscles were retracted to expose the lateral aspects of the right L2 through L5 vertebrae, as well as the dorsal surfaces of the medial transverse processes. Accessory processes from L2 to L5 were excised using 1 mm Friedman bone rongeurs (Fine Science Tools). These rongeurs were also used to remove portions of the laminar bone, exposing the distal third of the dorsal root ganglia (DRG). Fascia overlying the DRG was delicately removed with 0.1 mm ultra-fine clipper scissors (Fine Science Tools, catalog number: 15300-00). Animals were then positioned in spinal clamps to allow for precise DRG injections.

We verified the transduction rate of DRG neurons by computing the fraction of all neuronal cells (labeled with Nissl) that also expressed mCherry (our virally transduced cells’ tracer; mCherry was amplified by DsRed staining; Figure 2). The transduction rate of DRG neurons in a sample of *n* = 7 DRGs from 5 rats was found to be 41 ± 7% (mean ± SD), in agreement with our past study that used the same viral construct (44%; [26]). The protocol of the hemi-section study was to perform unilateral intraganglionic injections at segments L2–L5 and inject 1 µL of AAV2-hSyn-mCherry or AAV2-hSyn-hM3Dq-mCherry. A partial laminectomy was performed at segments T9–T11 followed by a complete hemi-section of the right hemicord.

### 2.5. Contusion SCI

Animals received a T10 level contusion at 200 kDyn impact force (Infinite Horizon Impactor, Precision Systems and Instrumentation). Measured impact force was not significantly different between groups (*T*-test; *p* = 0.21; total = 18, n = 11 experimental, n = 7 control) (Figure 3).

### 2.6. Aftercare

Post DRG injection surgery rodents received injectable Rimadyl, followed by ½ tab of Rimadyl once daily for three days. Subjects received a subcutaneous injection of saline on the day of surgery. Contusion injury animals received buprenorphine analgesia followed by Rimadyl post-operatively as with DRG injection animals. Additionally, contusion surgery animals required twice daily checks where bladders were manually expressed for up to two weeks following surgery, until bladder control was typically restored. Contusion animals also received Diet-Gel to enhance food consumption.

### 2.7. Behavioral Assessment

Rodents underwent assessments using the BBB scoring system to exclude animals whose contusion injury was outside the expected range of 5–8. A score that was notably higher indicated an imprecise impact with the Infinite Horizon impactor. Six of the rodents had significantly higher BBB scores following injury (range: 9.5–20) and/or peak impact forces above 250 kDyn, and were thus excluded. Graphs comparing BBB scores and contusion (kDyn) force amounts are provided in Figure 3. Control and experimental groups were not significantly different in BBB score or contusion force using a standard *t*-test.

### 2.8. Exercise Training

Treadmill training was implemented three times per week and structured to progressively challenge locomotor ability across five defined speeds: 16, 20, 24, 28, and 32 cm/s. These speeds were chosen for the primary reason that the severity of injury makes trotting/running speeds difficult for the rats in early weeks. Most animals can do very limited exercise during week one and begin to regularly step and bear weight by week three. Slower speeds were chosen to better match the rat’s own capabilities. Each of the training sessions spanned 30 min, during which the animals engaged in five-minute locomotor bouts at each speed. These intervals were interspersed with a minimum of one full minute to allow for recovery and minimize fatigue. Neuromodulation was applied by activating the excitatory DREADDs through administration of clozapine-*N*-oxide (CNO) intraperitoneally at a dosage of 2 mg/kg, 30 min prior to each session. Past research has found that a CNO dose of 2.0 mg/kg (e.g., injection volume calculated and adjusted each time for the animal’s weight during the study) provides effective activation of the DREADDs receptor over a two-hour time span while reducing the possibility of off target effects [32,33,34].

All of the training sessions were conducted on a five-lane motorized rat treadmill (Panlab LE8710RTS, Harvard Apparatus, Cambridge, MA, USA), with individual lanes separated by transparent plexiglass dividers to prevent physical interaction while maintaining visual exposure. This regimen was maintained consistently over a nine-week period following the initial injury to support recovery and assess longitudinal training effects.

### 2.9. Kinematic Recordings

Three-dimensional kinematic recordings were obtained using a dual-camera system recording at 250 Hz. Rodent groups were randomly split into “A” and “B” with the researchers blinded to experimental group. Group “A” underwent MOCAP recordings on Monday while Group “B” underwent MOCAP recordings on Wednesday. Both groups received treadmill training on Friday and each group received treadmill training while other rodents were recorded. For example, on Monday, Group A underwent MOCAP recordings and Group B received treadmill training and vice versa on Wednesday. We did not gather MOCAP data on week five as this intermediate time point is typically stable in the recovery process. Calibration images of a set of LEGO bricks^®^ that spanned the volume of the treadmill were obtained at the start of each session to enable 3D reconstruction of tracked features from the two camera views [35]. Positive reinforcement and treadmill acclimation was adequate to achieve locomotion; thus, we placed 3D printed “blockers” over the treadmill’s electric shock grid and did not use it. Video capture was initiated when the animal maintained a central position on the moving treadmill belt for four seconds, indicating constant speed locomotion at the treadmill belt speed. Five trials were collected at each of the five treadmill speeds for later kinematic analysis.

### 2.10. Automatic 2D Marker Tracking Using DeepLabCut

We used the DeepLabCut (DLC, version: 2.1.8.2) software tools as described in Mathis et al. [36] and Nath et al. [37] to estimate the locations of the ASIS (anterior superior iliac spine), hip (greater trochanter), knee, ankle, and MTP (metatarsophalangeal; foot) joints in our videos (Figure 1B). The hindlimb joints were manually tracked and then iteratively refined from predictions to produce an 11,143-frame training dataset. Images were 2048 by 700 pixels, 95% of the data was used to train the ResNet-50-based model and 5% for validation [38,39] and a p-cutoff of 0.9 was used to gauge the effectiveness of joint marker estimation. The trained DLC model was then used to automatically track the rest of the videos. In total, 5388 four-second video dumps were analyzed, yielding ~5.3 M frames. A total of 21,559 strides were extracted and analyzed as described below. We used a conservative statistical approach in which we averaged the data down to one average stride time series per rat, treatment, speed, and week before submitting to analysis with a mixed effects model (described below).

### 2.11. Three-Dimensional Reconstruction and Kinematic Analyses

A custom Python code base was used to process the data from DLC output to parameters ready for statistical analyses [28]. Two-dimensional joint positions from each of the camera angles were combined with calibration matrices generated through DLTcal in MATLAB (version: R2025a) to reconstruct 3D joint positions. Strides were delineated as one complete cycle of right hindlimb movement, from one paw touchdown event to the next. Paw touchdown was estimated to occur at the anterior extremal value of the MTP (metatarsalphalangeal; i.e., paw) x-coordinate (farthest forward), where the x-coordinate was defined along the treadmill belt. The anterior extreme position of the MTP was used to identify paw liftoff. Each stride was divided into swing and stance phases using these events. The swing phase of kinematics spanned from the moment the right hind paw lifted off the treadmill to when it touched back down again, or toe-off to heel-strike. The stance phase lasted from heel-strike to the following toe-off.

Derived kinematic quantities were computed from the 3D coordinates and the paw on- and off-events. These included minimum and maximum joint heights during each stride and phase, and the standard deviation of these quantities as a measure of their variability and range of motion. Joint angles and their deviations during each stride were also computed. Stride cuts were used to interpolate the 3D coordinate data to 100 points per stride, and then hip, knee, and ankle joint angles were computed as described previously [21].

Standard definitions of basic stride parameters were used [40]. Duty factor was computed as the fraction of the stride cycle that a limb is on the ground. Step height was defined as the vertical lift of the paw during the swing phase from the minimum to maximum z coordinate. Step lateral range was defined as the maximum minus the minimum y-coordinate (lateral position, across the belt) of the paw throughout a stride. Minimum ankle height computes the minimum clearance above the ground during locomotion of the paw.

### 2.12. DREADD Activation in Late-Stage Recovery

In order to evaluate whether DREADD activation was necessary for sustaining locomotor improvements at chronic time points, we implemented a withdrawal experimental design during the final weeks of the study. In weeks 7 and 9, kinematic data were gathered on a separate day without CNO administration. In week eight, CNO was reintroduced when animals were run again for kinematic data collection. Thus, weeks 6, 7, 8, and 9 formed an ABAB design. A separate publication is in preparation detailing the neuronal circuit changes accompanying this work.

### 2.13. Statistical Analyses

We used linear mixed effects models for statistical analyses (as implemented in the R “nlme” library; [41]). These models are useful for their robustness in handling hierarchical, repeated measures data, including missing data points, that frequently occur in spinal cord injury studies due to pose estimation tracking accuracy or animal loss. The raw 2D DeepLabCut output was reconstructed to 3D using the Python (version: 3.7.9) pipeline [28] and a custom LEGO^®^ calibration object and the MATLAB software package TyDLT [35]. Strides were then cut and the stance and swing phases estimated from anterior and posterior extrema of the paw position in the fore-aft direction of motion on the treadmill (the x-axis; anterior = paw touchdown, posterior = liftoff). We then extracted stride averaged statistical features—the maximum, minimum, range, mean, and standard deviation—from the time series of each parameter. These stride-level metrics were then averaged across all strides within each unique combination of rat, time point, and speed, producing one representative value per condition per animal.

We used linear mixed-effects modeling to evaluate the influence of treatment, recovery stage, and speed on the kinematic outcomes. The fixed effects in the model included (1) treatment group, with two categories: experimental and control animals; (2) time point, covering 10 distinct stages from before injury (baseline) through 9 weeks post-injury; and (3) speed (five values, 16, 20, 24, 28, and 32 cm/s; covering a minimum speed walk to a slow trot). To account for repeated measures within subjects, each model included a random intercept for individual rats and nested within that a random intercept for week. The significance of each fixed effect and interaction was assessed using an ANOVA on the linear mixed effects model. Where treatment or interaction effects were significant, post hoc analyses were conducted at each time point using the Estimated Means method (R package “emmeans”) with Holm’s correction.

We chose this specific model structure after comparing several models with the Akaike information criterion and Bayesian information criterion. Our simplest model was that with random intercepts for animal only. We then also fit (1) a model with a random intercept for week nested inside rat, (2) a model with a random slope for week for each rat, (3) a model with a random intercept for each rat and AR(1) covariance structure across weeks, (4) a model with a random slope for speed within each rat, (5) a model with random slopes for both week and speed, (6) a model with a random slope for speed by rat and AR(1) covariance structure across weeks, and finally (7) a model with random slopes for both week and speed by rat and AR(1) covariance structure across weeks. We found that the first model with only the addition of a nested random intercept for week within rat performed best when evaluated by AIC and BIC. Typically, models with a random slope for week, or week and speed, were the second-best fits, but the difference was significant, ~150 to 350 for AIC, with the cluster of more sophisticated models hovering within about ~50 to 100 of each other. We believe this was because the models that could intuitively seem to be better fits, those having, e.g., a random effect slopes for speed and/or time and the AR(1) covariance structure on time, provide only marginal benefits that are outweighed by the additional parameters required, because the data are quite variable across time and less significantly correlated, and because changes across speed did not vary so significantly between animals.

Joint angles were averaged across strides at each percent stance bin for every unique rat-time-speed combination, yielding a single average time series per animal per condition. These series were then grouped by treatment category, and the mean and standard error of the mean (SEM) were calculated at each stance bin across rats. Statistical comparisons between DREADDS and control groups for the time series were performed using statistical parametric mapping (Python spm1d package; alpha = 0.05, two-tailed test), with significant differences at *p* < 0.05.

## 3. Results

Rodents in the study received a control mCherry-only virus (AAV2-hSyn-mCherry) or an experimental virus with hM3Dq DREADDs during DRG injections (Figure 1D). Transduction rates and expression levels were similar to our prior publication [21], as shown in Figure 1 and described in methods. The virus was allowed time to transfect (3 weeks) prior to the 200 kDyn T10 spinal contusion. The measured force of contusion was not significantly different between control and experimental groups (*p* = 0.21; n = 11 experimental, n = 7 control). Researchers carrying out the experiments were blinded to the subject groups.

### 3.1. Stride Parameters

Stride-related kinematic parameters demonstrated marked alterations over time following injury and recovery (Figure 4). As expected, increases in speed were accompanied by both longer strides and higher stride frequency. By week nine, both treated (DREADDs) and control animals exhibited significant increases in stride length relative to baseline, with the experimental group showing slightly longer strides at low speeds (Figure 4A; *p* = 0.035; treatment × time: *p* = 0.021). In contrast, stride frequency was generally reduced following recovery, although DREADDs animals showed a modest trend back toward baseline at intermediate speeds (Figure 4B; treatment: *p* = 0.020; treatment × speed: *p* = 0.013). Duty factor decreased with speed as expected across all groups, but treated animals displayed a lower duty factor at the final time point compared to baseline (week nine) (Figure 4C; treatment × time: *p* = 0.02; treatment × speed: 0.01). Step duration also declined slightly with treatment at intermediate speeds (Figure 4D; treatment: *p* = 0.020; treatment × speed: *p* < 0.001; treatment × time: 0.021), and stance duration in treated animals returned more closely to baseline across most speeds (Figure 4E; treatment: *p* = 0.022; treatment × time: *p* = 0.007; treatment × speed: *p* < 0.001), excluding a smaller trend at 16 cm/s. Swing duration remained prolonged in both groups compared to baseline (Figure 4F; time: *p* < 0.001), but treatment effects were not significant (treatment: *p* = 0.59).

Stepping parameters and foot clearance metrics further reflected treatment-specific patterns. Step height remained lower in control animals post-injury, whereas DREADDs animals displayed a significant increase at low to medium speeds over the recovery period (Figure 4G; treatment × time: *p* = 0.021; treatment × speed: *p* = 0.025). Lateral step range, perhaps indicative of trunk or balance compensation via paw circumduction, increased after injury (Figure 4H; time: *p* < 0.01), but was not altered by treatment (*p* = 0.15). Minimum ankle height, a measure of how low to the surface the ankle may be pressed during ankle flexion, was altered significantly by treatment and time; while all animals exhibited increases following injury, the experimental group showed lower minimums overall (Figure 4I; *p* < 0.05 for treatment, time, treatment × time and treatment × speed interactions). A lack of significance in post hoc testing at individual speeds between early and late time points within each treatment group may suggest that random effects absorb these differences.

Step duration was examined longitudinally at five treadmill speeds to capture recovery dynamics (Figure 5). At lower speeds (16 and 20 cm/s), DREADDs animals consistently exhibited longer step durations from week two through week six compared to the controls (Figure 5; treatment: *p* = 0.02; treatment × time: *p* = 0.021; time: *p* < 0.001; treatment × speed: *p* < 0.001). Withdrawal of CNO in week seven did not appear to cause significant changes, whereas withdrawal in week nine coincided with step duration returning towards baseline values.

Mean ankle height during swing was markedly reduced in both groups post-injury by week two but gradually returned to baseline over time (Figure 6). DREADDs animals showed a consistent depression of mean ankle height at intermediate recovery points (treatment: *p* = 0.018; treatment × time: *p* = 0.014; treatment × speed *p* = 0.017), although by weeks 7–9, ankle height had recovered. The DREADDs group showed a linear increase in ankle height across the late chronic phase. Although ankle height dropped slightly during DREADDs withdrawal (week seven), the effect was not statistically significant in that individual week (*t*-test on estimated marginal mean). In week nine, withdrawal coincided with mean ankle height in swing trending towards baseline values.

Mean ASIS height and mean ankle angle along with estimates of their variation (standard deviation across the stride) are seen in Figure 7 with the same variables replotted from the prior study in the unilateral hemi-section model [21]. In this work (with contusion injury and bilateral DRG injections and afferent excitation) we did not see statistically significant increase in mean ASIS height with DREADDs activation (treatment: *p* = 0.29; treat × time: *p* = 0.11), although a trend towards higher ASIS heights in treated animals at weeks 7 and 8 aligned with previous findings. ASIS height variability was notably higher in DREADDs animals (treatment: *p* = 0.026) and specifically at an early time point (week two; *p* = 0.015) in contrast to the prior study in which control animals trended towards higher variability. Both groups showed a sharp post-injury increase in mean ankle angle (i.e., increased flexion), with recovery over time (time: *p* < 0.001), however, the effect of treatment was not significant. Ankle angle variability showed no significant differences (treatment: *p* = 0.19), though the trend here was opposite to the prior study.

### 3.2. Joint Angles (Figure 8)

Knee joint angle waveforms were found to have differences between control and DREADDs animals in week nine at 24 cm/s, using statistical parametric mapping to identify regions of significance. Here, treated animals showed more highly flexed knee angles than controls. We note that we used a highly conservative strategy for analysis of only submitting one time series per condition (unique grouping of rat, treatment, time point, and speed); e.g., we averaged all strides for each rat in each condition down to one average stride, before submitting to analysis. Generally, we observe a trend towards treated animals being pushed farther from baseline at early times points (weeks 2 and 4) and then reducing, becoming more flexed than the controls in the final week, although when analyzed at the rat rather than stride level, these differences are not significant (Figure 8). Mean hip and knee angles did not show statistically significant differences with treatment (*p* = 0.20 and 0.08, respectively).

### 3.3. Withdrawal of CNO Activation in DREADDs Animals

Withdrawal of afferent excitation in week seven did not appear to cause significant changes overall. However, mean ankle height during swing in week seven did show a trend towards the more injured state of earlier weeks (Figure 6A,C,E). Withdrawal of CNO in week nine coincided with step duration (Figure 5) and mean ankle height in swing (Figure 6) returning towards baseline values.

## 4. Discussion

Stride-level analyses revealed speed-dependent treatment differences. At select low-mid speeds, experimental animals trended toward reduced step frequency with increased stride length relative to controls at matched treadmill speeds (Figure 4A,B), whereas at other speeds, the groups did not differ. These patterns could reflect tonic afferent excitation increasing net limb stiffness via reflex pathways, prompting longer, less frequent steps, or alternatively augmenting phase-appropriate output via segmental/CPG circuits [42,43]. The fact that stance duration trended toward shorter at specific speeds in both this study and the prior hemi-section study suggest afferent excitation results in these shorter, potentially stronger steps. Smaller effects of afferent excitation at higher speeds (Figure 5) are consistent with the hypothesis that faster locomotion relies on feedforward locomotor networks more (e.g., CPGs) and relies less on afferent input [44]. The observed prolonged step duration (Figure 5) at lower speeds could similarly reflect an increased reliance on afferent feedback at those speeds during early to mid-recovery when descending and intrinsic spinal drive may be less forceful. At higher speeds (24–32 cm/s), the treatment difference on step duration was reduced. Alternately, long step durations may reflect the system attempting to override tonic afferent excitation that fights transitions between stance and swing—but this would predict that the step durations would return closer to the controls in both weeks 7 and 9 with CNO withdrawal, and while the week nine data might support this, a reduction was not seen in week seven, so this is not a clean conclusion.

The origins of elevated step height in the afferent excited group may reflect similar processes to the shortened steps: increased monosynaptic drive could amplify positive feedback from sense organs to muscles as they contract, increasing flexion and raising of the paw. As above, the effect could also be due to increased afferent drive through the CPG impinging on muscles [45]. It is interesting to note that the mean ankle height during swing is lower in the acute, intermediate recovery weeks (Figure 6B,C), but that it recovers to baseline or above in the final week. Lower minimum ankle heights in experimental animals suggested greater ankle compression (Figure 4I). Mean ASIS height trended higher under afferent excitation (consistent in direction with prior work [21]) but was not significant. Finally, the more flexed knee angle at the stance-to-swing transition in week nine may reflect enhanced force generation in response to neuromodulation over time. The trend across angles of acute stage treatment showing extension but final stage flexion could suggest a delayed but potentially adaptive effect of tonic afferent excitation, where sustained exposure during early recovery may promote compensatory adjustments that support more efficient performance once the excitation is withdrawn.

### 4.1. CNO Withdrawal

The withdrawal of CNO during weeks 7 and 9 was designed to provide insight into the extent to which locomotor changes depended on afferent excitation versus natural recovery or internalized (“permanent”) plastic changes. In both withdrawal weeks, step duration did not differ significantly from controls (Figure 5), consistent with a gradual recovery trajectory rather than a withdrawal effect. Ankle height estimates shifted toward control levels at multiple speeds with withdrawal in week seven (Figure 6A,C,E), with significant effects of treatment and the interaction of treatment × time supporting this shift. In control animals, while some parameters changed significantly in week nine with the second withdrawal (step duration: Figure 5A,B, ankle height: Figure 6; ankle angle variation: Figure 7D), many did not, and an expected strong repeat effect in weeks 7 and 9 that would highlight a strong acute effect of DREADDs excitation at these time points was not apparent.

These mixed effects may indicate that while afferent excitation supports the emergence of improved motor output, it may also introduce destabilizing elements during ongoing locomotion that require compensatory mechanisms to override, or simply the withdrawal of stimulation lets the system function well at long time points. The results of this ABAB design that extend further in time than the prior study (9 weeks versus 8) suggest, in agreement with that study, that effects of the neuromodulation are being “handed off” to the spinal cord circuity over time.

### 4.2. Comparison with Prior Hemi-Section Study

In this study, we used a contusion model that involves impacting the spinal cord with a blunt instrument at the midline, causing a more clinically relevant injury that has bilateral effects. Our prior study used a hemi-section injury in which one side of the spinal cord is cut using a fine blade. As such, it may be that general stiffening of the leg with afferent excitation may support the statistically significant higher pelvic height in the prior study and non-significant trend in this study—with both legs affected, it is just harder to reliably raise the pelvis, but afferent excitation is “attempting” to do this in both cases. The increased pelvic instability of the bilateral contusion may limit the effectiveness of afferent excitation and be the source of the trends and some significant changes towards higher variability in some parameters in the contusion injury that was the opposite of lower variability for those parameters with treatment in the hemi-section study (Figure 7). It is also possible that the more severe bilateral contusion injury limits the ability of spinal circuitry to adaptively redistribute load across sides of the body, using, e.g., “relay” or “dog-leg” circuits [46].

### 4.3. Plasticity

The long-term recovery trajectories and delayed improvements observed in several kinematic parameters, especially following CNO withdrawal, suggest the possibility of lasting plastic changes in the motor control system. While minor regressions were noted during the first withdrawal period, certain features—such as ankle joint trajectories and step height—showed trends toward normalization even in the absence of neuromodulation at the final time point. One possible explanation is that sustained afferent drive during the active rehabilitation phase engaged adaptive mechanisms in spinal circuitry or heightened coupling to descending pathways; peripheral contributions such as muscle strengthening may also have played a role. Although speculative, these patterns are consistent with the idea that prolonged tonic excitation may condition intrinsic spinal networks or alter the ability of supraspinal commands to regulate sensory feedback, warranting further study in experiments.

### 4.4. Potential Confounds

It is important to note that there are several non-injury related factors that may also have influence locomotor outcomes over the 9-week study period. Rats were maturing from young adulthood toward middle age during this time frame, a developmental window associated with steady increases in body weight (~20% in female Long-Evans rats). Such weight gain could have effects on stride length and joint kinematics due to allometric scaling effects, but our daily calculated 2 mg/kg CNO dosage should ensure comparable effectiveness of our activator throughout the study. In addition, repeated treadmill exposure (three sessions per week for nine weeks) likely improved task familiarity, which thereby may have contributed to more efficient stepping patterns at later time points, although all animals were acclimated to the treadmill before the injury. Since we did not follow a sham-injury group longitudinally, we cannot exclude the possibility that some time-related changes in stepping parameters partly reflect growth, maturation, or practice effects. These caveats should be considered when interpreting the results.

### 4.5. Future Directions

To understand the contributions of monosynaptic afferent to muscle pathways versus polysynaptic pathways, future genetic experiments could attempt to restrict afferent excitation to these populations of neurons. Force plate data would be useful to determine whether afferent excitation is shortening stance duration through higher forces in the affected limbs; presumably this is the case, but it is possible that compensation arises from other postural changes or reliance on the forelimbs. Furthermore, extensive studies have analyzed and modeled how rodent gait is altered by spinal cord injury [47]. It would be insightful to extend these modeling studies to include injury with afferent neuromodulation, and potentially to include models of CPG circuitry [45].

## Figures and Tables

**Figure 2 bioengineering-12-01080-f002:**
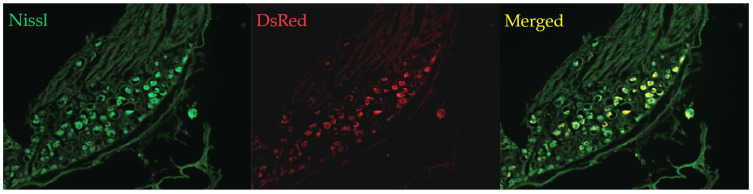
Images of Lumbar DRGs to show transduction rate. (**Left**) Nissl stain of DRG. (**Middle**) DsRed (DREADD-positive, transduced by our viral construct). (**Right**) Merged image showing doubly Nissl- and DREADD-positive cells (yellow). Verification of transduction rate by computing the ratio of red cells to green cells in 7 DRGs from 5 rats resulted in a 41% transduction rate (41 ± 7%). This agrees with a previous paper using the same viral showing a 44% transduction rate [21].

**Figure 3 bioengineering-12-01080-f003:**
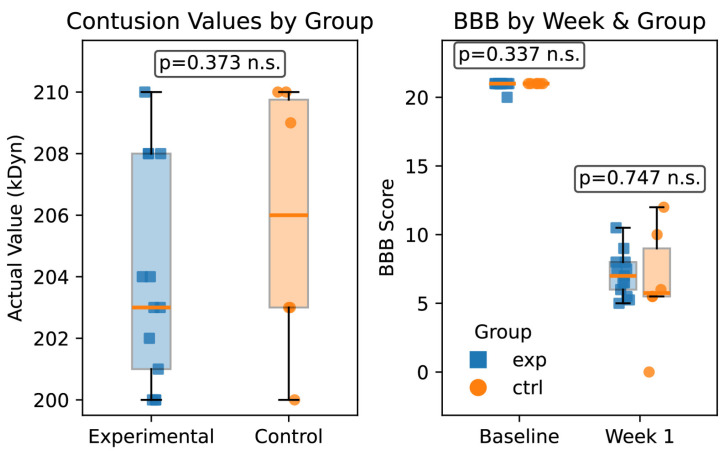
Box plots showing baseline behavioral assessment and contusion force values from Infinite Horizon Impactor. *T*-tests show non-significant (n.s.) differences between contusion values (**left**) and BBB scores (**right**) for both baseline assessments and week one post-operation.

**Figure 4 bioengineering-12-01080-f004:**
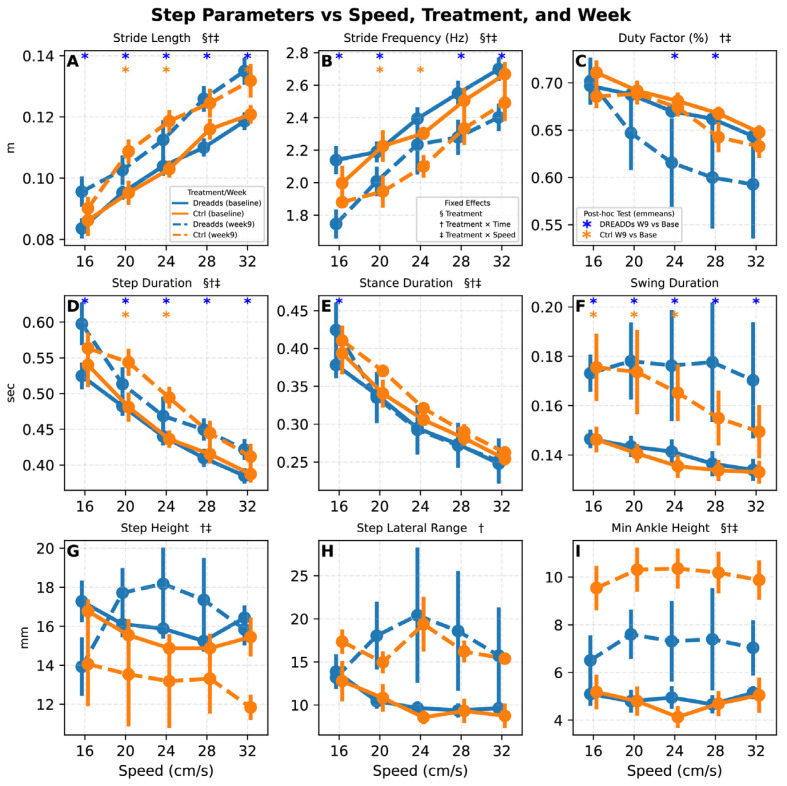
Stride parameters and three stepping metrics at baseline and the conclusion of the study versus speed and treatment. The §, †, ‡ symbols next to parameter names in the panel titles indicate significance of fixed effects for treatment (§), treatment × time (†), and treatment × speed (‡), respectively. Asterisks (*) denote significance of post-hoc testing on estimated marginal means. Animals increase speed with increases in both stride length and stride frequency [41]. (**A**) Both experimental (blue) and control (orange) animals show increased stride length after recovery in week nine; treated animals have slightly longer strides at low speeds after recovery (Linear Mixed-Effects Model; hereafter “LME”; treatment: *p* = 0.035, treatment × time point: *p* = 0.021; treatment × speed: *p* = 0.031; all further p values correspond to the linear mixed effects model with structure described in body text). (**B**) Conversely, all animals showed reduced stride frequency after recovery, but treated animals had a slight increase towards baseline at the intermediate speeds (treatment: *p* = 0.020; treatment × speed: *p* = 0.013). (**C**) Duty factor was reduced with treatment (treatment alone not significant: *p* = 0.41; treatment × time point: *p* = 0.018; treatment × speed: 0.01). (**D**) Step duration showed a slight decrease toward baseline with treatment at intermediate speeds (treatment: *p* = 0.020; treatment × speed: *p* < 0.001; treatment × time: 0.021), while (**E**) stance duration was notably back to shorter baseline values with treatment for all speeds except a smaller trend at 16 cm/s (treatment: *p* = 0.022; treatment × time: *p* = 0.007; treatment × speed: *p* < 0.001). (**F**) Both experimental and control groups exhibited notably slower swing durations than their baseline counterparts (time point: *p* < 0.001), although treatment: (*p* = 0.59) and treatment × time point (*p* = 0.64) were not significant. (**G**) Injured control animals showed a reduced ability to lift their paws to the original baseline height while experimental animals showed a notable increase in step height at low to medium speeds (treatment × time point interaction: *p* = 0.021; treatment × speed: *p* = 0.025). (**H**) Step lateral range increased with injury and recovery (time point: *p* < 0.01), but was not altered by treatment (*p* = 0.15). (**I**) Minimum ankle height, a measure of how low to the surface the ankle joint got during a stride, was significantly increased for both treatment groups with injury and recovery, but experimental animals exhibit a lower minimum height (*p* < 0.05 for treatment, time, treatment × time, and treatment × speed).

**Figure 5 bioengineering-12-01080-f005:**
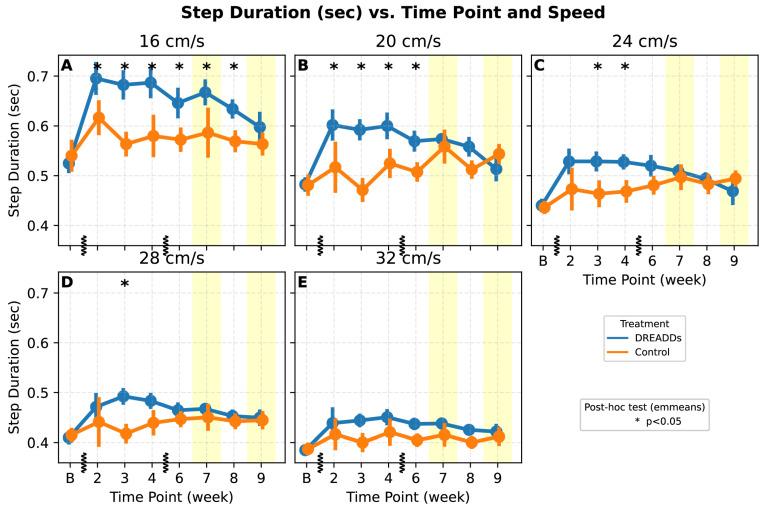
Step duration across recovery week at five varying treadmill speeds. Vertical hashes on the x-axis denote breaks from linear timeline. At lower speeds (16 and 20 cm/s), DREADDs animals (blue) consistently exhibit prolonged step durations when compared to controls (orange), especially during early to mid-recovery (LME: treatment *p* = 0.02; treatment × time: *p* = 0.021; treatment × speed: *p* < 0.001; time: *p* < 001; asterisks: *t*-test on estimated marginal means from the LME). Yellow shaded regions denote weeks seven and nine in which DREADDs activation was withdrawn during the kinematic data capture, and “B” denotes the baseline time point. (**A**) DREADDs animals show significantly prolonged step durations compared to controls from week two through week six, with a return toward baseline by week nine. (**B**) A similar pattern emerges in 20 cm/s when compared to 16 cm/s: DREADDs animals exhibit consistently longer step durations than controls through week six. (**C**–**E**) Differences persist in moderate speeds and high speeds but are slightly less pronounced.

**Figure 6 bioengineering-12-01080-f006:**
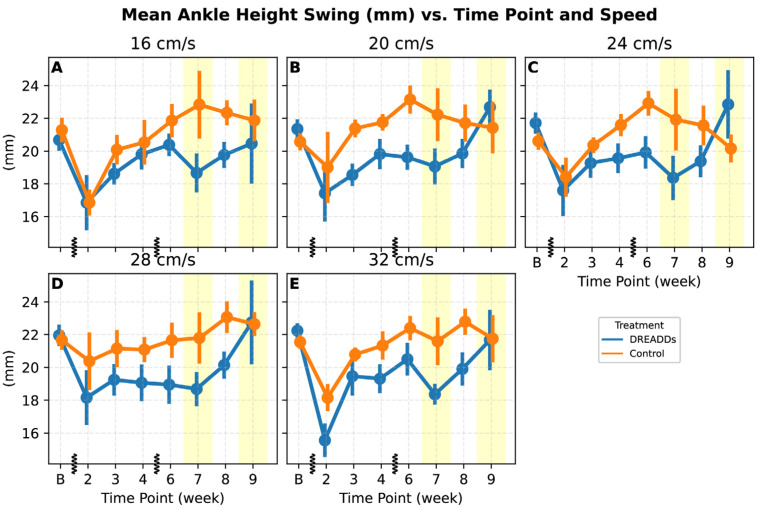
Mean ankle height during the swing phase versus recovery time and speed for both DREADDs (blue) and control (orange). Vertical hashes on the x-axis represent breaks in the linear timeline. Yellow shaded regions denote weeks seven and nine with withdrawal of DREADDs activation during kinematic data collection. (**A**–**E**) Ankle height drops precipitously from baseline to week two in both groups following injury and then recovers over time. Across speeds, DREADDs activation appears to suppress the ankle height during swing at intermediate time points, but in final weeks with DREADDs withdrawal this parameter returned to baseline (LME; treatment: *p* = 0.018; treatment × time: *p* = 0.014; treatment × speed: *p* = 0.017). Both groups show relatively flat trajectories at higher speeds, with a mild overshoot of baseline by the control group. Interestingly, DREADDs animals show linearly increasing height in chronic phase weeks 7–9. Withdrawal of DREADDs activation in week seven may have caused slight regression of ankle height (**A**,**C**,**E**), but this effect was not significant in post hoc testing (*t*-test on estimated marginal means).

**Figure 7 bioengineering-12-01080-f007:**
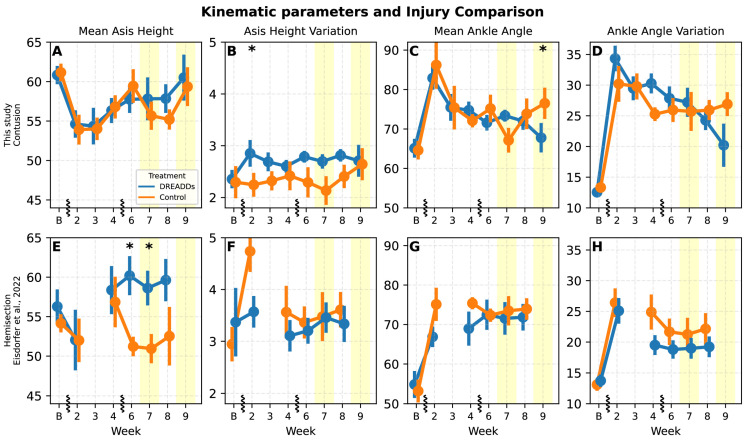
Kinematic parameters for comparison with a prior unilateral study employing the hemi-section injury model [21]. In that study, a right T9 level hemi-section injury was utilized with unilateral right-side transduction of L2–L5 DRG afferents. Vertical hashes on the x-axis denote breaks in the linear timeline (week) and yellow shaded regions to represent weeks with CNO withdrawal (weeks seven and nine). Asterisks (*) denote significance of post-hoc t-tests carried out on estimated marginal means. (**A**,**E**) In this study, we do not see the statistically significant increases in mean ASIS height with DREADDs activation that were seen in the prior work in the chronic weeks six and seven (speed 24 cm/s; this study: LME: treatment *p* = 0.29; treatment × time *p* = 0.11); although a trend to higher values for DREADDs activation is seen in weeks seven and eight, that agrees with that prior result. The interaction of treatment with speed was significant (*p* = 0.012). (**B**,**F**) ASIS height variation (standard deviation across a stride) in DREADDs animals was increased (LME: treatment: *p* = 0.026); significant at early time point (weeks two; *p* = 0.015; *t*-test on estimated marginal means). This was opposite to the trend seen in the hemi-section study of lower variability for treated animals. (**C**,**G**) In this study, both DREADDs and control animals showed a sharp increase in mean ankle angle following injury (increased flexion; LME: time *p* < 0.001), and subsequent recovery to near baseline, but the effects for treatment were not significant. In contrast, in the hemi-section study, DREADDs animals exhibited a trend towards more flexed ankle angle in the early to middle time points (weeks one to four). (**D**,**H**) The main effect of treatment was not significant for ankle angle variation (standard deviation) in this study (LME: treatment *p* = 0.19). It showed a trend towards being higher than controls in intermediate weeks before converging, whereas in the prior study the trend was for treated to be lower throughout; the same pattern in ASIS height variation. (**E**–**H**) data re-plotted from those gathered for Eisdorfer et al., 2022 [21].

**Figure 8 bioengineering-12-01080-f008:**
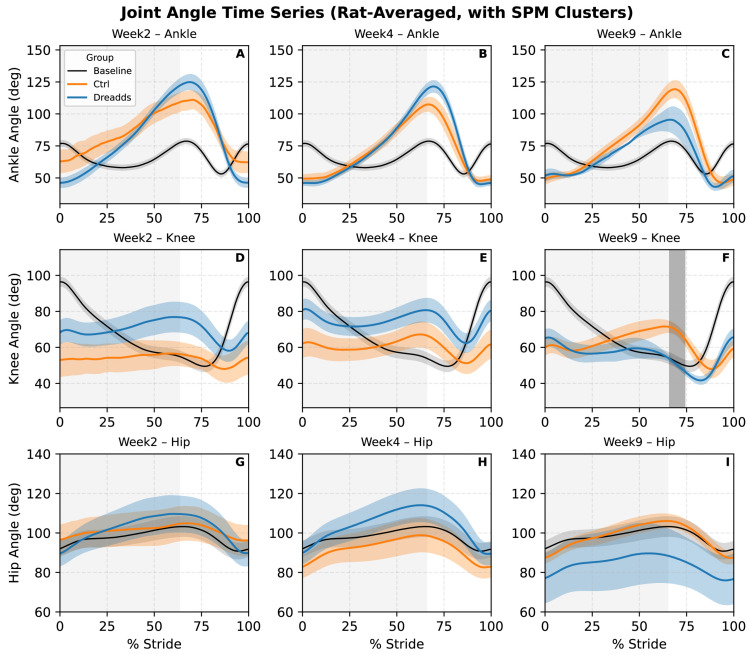
Joint angles at early (week two), intermediate (week four), and final (week nine) time points at 24 cm/s, right column. Each panel shows mean joint angle versus percent stride for baseline (black/gray), control (orange), and DREADDs animals (blue). All strides were averaged to one time series per rat, treatment, time point, and speed before producing these figures to be statistically conservative; thus the unit of measure is rat, not stride. Shaded regions represent standard error of the mean (SEM). Panels (**A**–**C**) show ankle again over percent stride immediately after injury (**A**), at the transition between acute phase to chronic phase (**B**), and at the final time point (**C**). Panels (**D**–**E**) show the analogous data for the knee joint, and panels (**G**–**I**) for the hip joint. Vertical gray patch in (**F**) denotes a region of statistical significance between the control and DREADDs groups (statistical parametric mapping: *p* < 0.05), and the light gray shaded region denotes stance phase. The central peak in ankle angle corresponding to maximal plantarflexion (extension for push-off), and the following drop indicates flexion for swing. Injury causes a more extended ankle for much of the stride, and especially throughout the stance to swing transition (push and liftoff), followed by an overly flexed ankle at touchdown and initial stance phase. At nine weeks (**F**) DREADDs animals exhibited increased knee flexion during push-off when compared to controls.

## Data Availability

All data supporting the conclusions of this article will be made available upon request to the authors.

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
