# Peer review of "Detailed Kinematic Analysis Reveals Subtleties of Recovery from Contusion Injury in the Rat Model with DREADDs Afferent Neuromodulation"

_bioengineering, 2025, doi:10.3390/bioengineering12101080_

Round 1

Reviewer 1 Report

Comments and Suggestions for Authors

1. The introduction should include a broader discussion of rehabilitation approaches and their impact on recovery; relevant recent work (e.g., Ageeva et al., 2024, DOI: 10.3390/ijms25073772) may be cited.

2. Subsection 2.1 (Experimental Design) provides general information that is repeated later; consider removing or merging.

3. The starting point of treadmill training after injury is not clearly stated.

4. It is not indicated whether neurological status was assessed prior to training (e.g., BBB score).

5. Clarification is needed on whether animals were able to support hindlimb weight at early stages, whether external support was required, and how variable this ability was across subjects. These details are necessary to evaluate the adequacy of the rehabilitation procedures.
6. While LME with ANOVA and Holm-corrected Wilcoxon is appropriate for scalar metrics, joint-angle waveforms were tested point-by-point (100 bins per cycle) with uncorrected t-tests at p<0.05. This inflates false positives. Methods such as Statistical Parametric Mapping, cluster-based permutation, GAMMs, or strict FWER/FDR correction should be applied to ensure reliability of the reported “significance bars.”

7. Only random intercepts per animal were included. Given repeated measures across speeds and weeks, random slopes and an appropriate covariance structure (e.g. AR(1) over time) should be considered to better reflect the data hierarchy.

8. Although LME is robust to missing data, transparency is needed. With 5.3M tracked frames, the proportion of excluded strides/frames per group and week (p-cutoff 0.9) should be reported, along with the number of strides analyzed per condition (median [IQR]). Without this, observed variability (e.g. duty factor) may be confounded with tracking variability or data imbalance.

6. In the section of results comparing the contusion model with the previous hemisection study (Eisdorfer et al., 2022), the similarities in methods and analyses are substantial. Although the present work introduces a more clinically relevant model, longer follow-up, and 3D kinematics, the novelty should be articulated more explicitly to avoid the impression of fragmented publication.

7. Claims of “resistance-like training” and “bounce-back recovery” are overstated. These ideas remain hypotheses not consistently supported by the data. The Discussion should temper this language or provide stronger evidence.

8. Statements about reduced treatment effects at higher treadmill speeds are qualitative. A formal treatment × speed interaction analysis should be reported to support this claim.

9. Comparison with prior work: The link to the previous hemisection study (ref. 15) lacks quantification of viral expression and does not clearly highlight novelty. Data on transduction rates should be provided, and the distinct contributions of this contusion model (bilateral injury, longer recovery, 3D kinematics) emphasized.

10. Interpretation of weeks 7 and 9 is ambiguous. The Discussion should distinguish natural time-related recovery from pharmacological withdrawal effects more explicitly.

11. Explanations involving reflex-driven stiffness or increased force lack EMG or force data. These should be framed as hypotheses rather than conclusions.

Reviewer 2 Report

Comments and Suggestions for Authors

Gavin Thomas Koma, Kathy Keefe, George Moukarzel, Hannah Sobotka-Briner, Bradley C. Rauscher, Julia Capaldi, Jie Chen, Thomas J. Campione 3rd, Jacquelynn Rajavong, Kaitlyn Rauscher, Benjamin D. Robertson, George M. Smith, Andrew Spence

Detailed kinematic analysis reveals subtleties of recovery from contusion injury in the rat model with DREADDs afferent neuromodulation.

COMMENTS FOR THE AUTHOR:

Spinal cord injury (SCI) is a serious disease that is widespread in the world. The causes of such injuries are different, so the methods of treatment are also different. There is no doubt about the relevance of searching for methods of treatment and rehabilitation of patients with spinal cord injury. In many cases, SCI causes a complete or partial disruption of the nerve canal, and as a result, the control of motor functions by the central nervous system is impaired. The sensorimotor neural network is damaged, which leads to loss of function with various etiologies in the acute and chronic phases of SCI. Recently, a technique of epidural electrical stimulation has been developed, which gives promising results. In fact, the technique is based on the search for "sleeping" neural networks that can be activated by electrical stimulation (the effect of erosion). This opportunity appears due to the presence of a large number of neurons and processes, the connections between which can be strengthened by stimulation (Hebb synapse).

The authors of this paper seek to enhance the effects of epidural electrical stimulation by administering specific substances that could increase the excitability of neural network components. In this study, they aimed to characterize in detail the effects of chemogenetic stimulation of large-diameter peripheral afferents on the kinematics of locomotor recovery in a rat model of SCI contusion. Rats had afferent activity stimulated by excitatory designer receptors activated exclusively by designer substances (DREADDs) targeting large-diameter dorsal root ganglia neurons. The authors showed that DREADDs-treated animals exhibited a “bounce” recovery pattern characterized by a greater deviation from baseline than controls in step length, duty cycle, step height, and ankle angle during the acute phase, which then recovered to baseline during the final week after cessation of neuromodulation. The long-term recovery trajectories and delayed improvements observed in several kinematic parameters, especially after CNO withdrawal, indicate the occurrence of persistent plastic changes in the motor control system. It is expected that it will be possible to identify those neuronal populations that are most susceptible or suitable for genetic experiments.

The article contains an abstract, introduction, sections “Materials and Methods”, “Results”, “Discussion”, an analogue of the conclusion, and literature.

  1. The title fully reflects the content of the review.

  2. Abstract is it really a summary, include key findings and have an appropriate length.

    4. This study and its introduction provide a complete picture of a new methodological approach to treatment after spinal cord injury.

  1. 5. Final comments.

I recommend publishing the review unchanged.

Reviewer 3 Report

Comments and Suggestions for Authors

This paper uses a novel approach in an ambitious attempt to better understand the effects of afferent excitation on motor recovery after SCI.  The investigators appropriately blinded the study and did a painstaking analysis of many components of stepping.  While this is a generally well designed study, the manuscript suffers from 3 common writing problems that make the paper difficult to follow and the data not very useful to other investigators.  The manuscript will require significant rewriting if the authors want others to actually read it and make use of the information. The authors need to rethink how they explain the rationale for their experimental plan and how they organize the presentation of their data and its interpretation.  

 Rather than list specific issues in the order that they arise in the manuscript, I have organized by theme so that the authors can better understand how to make changes that will enhance clarity and make their paper suitable for publication.  

  • Failure to recognize that readers do not know everything that the authors know.

Activation of neurons with DREADDs is not a common methodology.  The level of the contusion injury is lower than is typical and uses a different rat strain than is used in most studies, and the methodology for assessing changes in locomotor function is unique to this group.  However, the authors make little attempt to explain how their DREADDs strategy selectively activates large diameter sensory afferents or validate that their protocol worked as intended. 

1a) The abstract (as written) would seem to imply that DREADDs selectively affected sensory afferents, whereas the selectivity was a function of the viral vector used and direct injection into DRGs.  Likewise, it is not clear that CNO is the designer drug that is being referred to in the term DREADDs (Designer Receptors Exclusively Activated by Designer Drugs). This is only explained part way through the methods. 

1b) The authors never say what proportion of the DRG neurons were transfected with the DREADDs, only that it was similar to their previous study.  Since knowing the approximate percentage of transfected cells and the specificity is critical to interpreting the results, this omission would force the reader to stop and read another paper before assessing the data.  (The 44% indicated in the previous paper is a good enough number to expect substantial activation of the targeted population.  5% would not and 95% would indicate potential oversaturation of targeted cells.)

1c) The authors never say anything about the pharmacokinetics of CLO or provide any evidence that the dosing was sufficient to keep the neurons activated for the full 30min timepoint of analysis.  Based on a cursory reading of the literature, levels of CLO that the DRG were exposed to probably did not peak until the end of the evaluation period but likely were high enough throughout the evaluation period to be effective and continued to activate neurons for another couple of hours, but this is only a guess.  Unlike electrical stimulation, chemical stimulation cannot be turned on and off to correspond with treadmill training periods and it is not clear how neuronal stimulation that is not linked to locomotion may affect the response.  

It would have been simple enough to include this critical information in the Introduction in a short paragraph stating (1) generally how DREADDs activation worked, (2) that other investigators had successfully used the vector the authors were using to selectively transfect neurons with DREADDs, and (3) that a previous paper by the authors validated that approximately 40% of DRG neurons get transfected with the methods used in this paper and that the dosing of CLO was found to be sufficient to have an effect on locomotor recovery in a different model of SCI. This would have left some questions about the duration of the neuronal activation for the discussion but would at least have cleared up questions about whether the basic protocol worked as intended.

1d) The authors do not describe the severity of the injury or extent of recovery in a way that allows for comparison to other injury models.  The B-B-B (Basso-Beattie-Bresnahan) scoring system has become the standard for describing effects of CNS or PNS injury on hindlimb locomotor function.  Typically, baseline B-B-B scores or at least a general description of the impairment or a photograph of the hindlimb stance is included in any study of locomotor recovery, even if the primary focus of the paper is a more detailed kinematic analysis of motion.  In the author’s previous publication, they included sideview photographs of an animal which clearly showed several of the key characteristics for ranking impairment.  I could easily compare this injury to other models.  (Some of us have seen a lot of rats walk and recognize characteristic features of the posture and stance.) The authors also helpfully marked the 5 key points that they were tracking in the previous paper, which allowed a visualization of the kinds of angles they were looking at. Such an image would be very useful in this paper as well.

1e) The authors indicated the numbers of images and lengths of videoclips of locomotion used for analysis, but not whether these were from the first, second or third day of training in the week, or a mix of all three.  They also did not say if the same number of steps were analyzed for each animal, or whether the samples may have been skewed to represent only a few members of each group that were easier to analyze.  The standard deviations for some of the measurements are quite large.  Does this represent animal to animal variability or do individual animals step erratically?

1f) The authors indicate that 30 animals were used, but not the n values for each group.  I presume that there were 15 animals per group, but some animals may have been euthanized due to post surgical complications, or the authors may have used larger numbers of animals in the treatment group.

1g) Abbreviations that are not in widespread use in papers relating to models of spinal cord injury (such as SCI and DRG), need to be spelled out on first use. 

DREADDs (Designer Receptors Exclusively Activated by Designer Drugs) is used in both the abstract and only spelled out in key words and list of abbreviations. (Somone just reading the abstract would have no idea what the manuscript is about.)

Similarly, ASIS (anterior superior iliac spine) or iliac crest is only spelled out in the list of abbreviations, as are LME. MCP, and MTP.  Readers need to know what you are talking about and not search for a list of abbreviations and lose their place in the manuscript

1h) References are often cited in ways that make it unclear what the cited papers actually show. 

For example, line 64-66: “Studies have shown that EES targeting the dorsal roots of the lumbosacral segments promotes motor recovery and overall locomotion in animal studies and human SCI cases [6, 9-14].” would appear to indicate that all referenced papers included both animal and human studies.

Likewise, line 89-90: “In past work, we observed changes in kinematics that likely reflected these differences [15].” coming immediately after a paper about epidural stimulation, appears to indicate that the author’s previous paper also looked at epidural stimulation in human SCI patients.

Similarly, line 72-74: “Unfortunately, with electrical epidural stimulation, it is not precisely known which neurons are being recruited during stimulation, and thus recent studies are bringing genetically encoded activating tools to bear on the problem [9, 15].” One paper used electrical stimulation in mice and humans and the other used genetically encoded activating tools in rats. One cannot get to the conclusion from the facts presented in these papers without making a lot of assumptions.

1i) Since much of the design of the experiments in this study are based on the protocols and results of the author’s pervious study, readers need more detail about those methods and findings than just “We utilized a similar paradigm but with a hemi-section injury model in our prior work [15]” or “In past work, we observed changes in kinematics that likely reflected these differences [15].       

1j) The duration of the study and the speeds chosen for treadmill training seem arbitrary as no rationale is given for them.  At 9 weeks, spontaneous recovery for most rat SCI models has plateaued and therefore this endpoint seems reasonable, but I am not really sure where the other speeds come from.  As near as I can tell, these are all speeds where the normal gait of a rat would be expected to be a walk, rather than a trot.  If the authors intended to evaluate parameters of leg movements across a range of walking speeds, without forcing a transition to a trotting gait, the numbers make sense.  If they were matching treatment durations and/or walking speed tests commonly used in EES studies of rodents, this would also be a good reason.

1k) There is no explanation for the lack of motion capture data for week 1,2, and 5.  If the animals were not capable of weight bearing stepping on week 1 and 2, say so.  If there was some reason why recordings could not be made on week 5, tell the reader. 

  • Organizing and describing data based on the authors’ own focus of interest rather than providing the necessary context to understand the full data.

2a) The second sentence of the abstract talks about epidural stimulation as a promising new treatment for SCI, implying that this manuscript will focus on epidural stimulation, or at least compare epidural stimulation to some other treatment. Then the authors seem to abruptly change topic to say that the authors will therefore focus on chemogenetic stimulation.  Then they say that they will use DREADs, without ever making it clear how this will work.  Do the authors not understand why this is confusing?

2b) For no apparent reason, the authors begin by comparing the baseline data to the 9 week timepoint and then look at a time course.  It would seem more logical to show the time course of changes in locomotion and then compare the baseline to the endpoint.

2c) Authors absurdly refer to some of the parameters that they measure as “gait parameters” and others as “stepping parameters”.  An animal’s gait refers to the pattern of coordination of limbs, not some portion of the movement of an individual limb.  Some parameters of locomotion of an individual limb can be affected by a change in gait, but that does not mean that they are gait parameters.  Since what the authors actually measured were components of stepping of a single limb, they were all stepping parameters.

2d) Titling a section and figure: ” 3.2. Comparison with hemi-section injury model (Fig. 4)” makes no sense.  The authors do not show any comparison between data from this and the previous model and they only show a subset of data that they also show for the previous model.  Furthermore, all discussions of comparison between data from this study and their previous study belongs in the Discussion.

2e) Headings of Figures do not reflect their contents.  Figure 3 is not a 3 dimensional plot. Just get rid of these heading as they just confuse things.

2f) There is no apparent logic behind what data is shown and what is not, or how data is grouped. Authors show plots with minimum ankle height and iliac crest heights in separate figure and do not show knee height at all.   Why are ankle height and angle import measures, but not knee?  Section 3.3.  is labeled, “Joint Angles” but it only talks about the ankle joint.   There are 2 other joints in the leg.

2g) The terms “buckling” and “bounce back” seem to have been made up by the authors to characterize phenomena that are never properly identified here.  Bucking is described in the previous paper as a collapse of the hindquarters (specifically associated with withdrawal of CNO), but authors just seem to assume that readers understand what they are talking about.

2h) It is not clear reason why the authors chose to show the changes in ankle angle over time for week 4 and week 9 (fig 6), rather than say week 4 and week 8, since week 9 did not have CNO injections and this would confound the effects of increased time with treatment withdrawal.  Why is there no baseline comparison to show how the movement changes after injury compared to an intact animal? 

2i) In general, there is a lot of explanation and interpretation of the data which is only presented in figure legends and really belongs in the Results or Discussion. The figure legends should just describe the figures.

2j) In both the Results and Discussion, authors frequently refer to something changing, but then also say the difference was not significant.  For example, line 4-4-415: “Though not significant, mean ASIS height was elevated under afferent excitation.”  line 427-428: “Similarly, while ankle height regressed toward control levels strongly at multiple speeds (Fig. 4), these did not rise to statistical significance.”  If differences are not statistically significant, they are not different.  The most that one can say is that there is a trend towards an increase or decrease.

2k) Statements are made in the Discussion that are not entirely accurate.  For example: Line 388-389 “Specifically, animals in the experimental group demonstrated reduced step frequency coupled with increased stride length at comparable speeds (e.g., Fig. 2AB).”  Reduced stride frequency and longer strides were only observed for some speeds.

2l) Graphs which plot values over time should be presented such that the distance between points corresponds to the time difference.  In figure 3 and 4 weeks 1,2,and 5 are missing and data is plotted as though equal amounts of time elapsed between measurements.  This can distort perception by make gradual changes appear more abrupt.

  • Failure to address potential effects factors not being evaluated on outcomes.

In addition to the contusive injury and recovery from SCI, 3 other things change over the 9 week period of the study.  The animals (1) get older, (2) gain weight, and (3) gain practice walking on the treadmill.  For a rat, 9 weeks is a substantial portion of their lifespan and starting at the young adult stage, it represent a period of maturation comparable to the start of a transition to middle age.  Based on the reported growth curves, female Long Evans rats would be expected to increase their body mass by about 50g, or 20% of their starting weight. Increased body weight could not only alter stride length and other parameters of locomotion, but would also result in a lower concentration of CNO if the same dose is delivered throughout the study.  Three treadmill sessions per week for 9 weeks would give animals a lot of opportunity to get used to adjusting their walking to the forced control of their speed.  Since there were no sham surgery animals to follow over time, the authors cannot rule out the possibility that the increase in stride length for both the DREADDs and control animals, for example, was not due in part or whole to the animals just getting a bit bigger, older, and more experienced in how to most comfortably walk on a moving treadmill.   It is not necessary to repeat all of these measurements with unoperated or sham contusion animals, but it is necessary to be aware that some of the changes observed over time may not be due to how the animals recover from SCI, but could be just a function of elapsed time.

Minor points:

Graphs in figure 2 are weird.  Markers for control and DREADDs groups do not line up and there are no indications of statistical significance between baseline and 9 week post injury timepoints or between control and DREADs group for any parameter, although the error bars suggest that some are different.

Markers also do not line up in figure 4 and I think that figure 4C and 4D were meant to be labeled as ankle angle and ankle angle variation.

Figure 1 shows that no motion capture was performed on week 2, but figure 3 and 4 show data for week 2.

Round 2

Reviewer 1 Report

Comments and Suggestions for Authors

I thank the authors for the work to improve the article. It can be accepted for publication in the present form.

Reviewer 3 Report

Comments and Suggestions for Authors

This revised manuscript is a vast improvement compared to the original submission.  It is now much clearer what the authors did, how they did it, and why they made the choices they did.  Their injury model and evaluation of locomotor recovery can now be compared to other lesion models and other analysis methods because they have provided the kinds of information necessary for making those comparisons.

There are a couple of remaining significant issues and a few minor type-Os, missing punctuation, and some confusing wording.

Issue 1: Figure 6 has an inset showing symbols used for specific levels of statistical significance for specific types of comparisons.  However, the actual comparisons are not clear and there are no such symbols on the graphs themselves for this figure.  Similarly, one of the symbols is used in figure 5, and these symbols are used in figure 4, without a symbol key.  The figure legend for figure 6 lists statistically significant comparisons that are indicated on any of the graphs.  Symbols should not be used in figure titles. This is just sloppy data presentation.

Issue 2: It is not appropriate to include data in a manuscript for which methods are not described or which have been previously published. Figure 7E-H should therefore be removed and a description of these results placed in the discussion.  … Unlike the results for our previous hemisection injury model which showed …

Line 94: I’m not sure what “scientific cutting” power is supposed to mean.

Line 130-131: missing comas “We utilized a similar paradigm, but with a hemi-section injury model, in our prior work [21], and thus a secondary goal of this work was to compare ….”

Line 418: clarify wording ….  increase in step height  (no increase in depression of step height)

Line 462: I think that you mean to say that wave forms for DREADD and control animals differed

Line 469: “when analyzed at the rat rather than stride level” is confusing. I think that you mean when analyzed for individual animals rather than for all animals in each treatment groups.

Line 474-475: This is confusing “Withdrawal of afferent excitation during kinematic recording in week seven did not introduce statistically significant changes in the data. Mean ankle height during swing showed a trend back towards the injured state (Fig. 6ACE).” 

Of course, adding new data would not change the data.  Do you mean that it did not change measured parameters compared to the previous week? Animals are still in an injured state.  Do you mean that mean ankle height in DREADD animals moved towards the values for the control treated group?

Line 474-478:  In section 3.4, Readers are required to go backwards to look at 2 previous figures.  It would be better to make these points when the figures were first mentioned.  It gets confusion to go back and revisit data that was previously analyzed. 

Line 483: missing coma : “…whereas at other speeds, the groups did not differ.”

Line 502: As above, not as before.

Line 506: There is no figure 1I.

Author Response

Reviewer 3

This revised manuscript is a vast improvement compared to the original submission. It is now much clearer what the authors did, how they did it, and why they made the choices they did. Their injury model and evaluation of locomotor recovery can now be compared to other lesion models and other analysis methods because they have provided the kinds of information necessary for making those comparisons.

We thank the reviewer for these comments.

There are a couple of remaining significant issues and a few minor type-Os, missing punctuation, and some confusing wording.

Issue 1: Figure 6 has an inset showing symbols used for specific levels of statistical significance for specific types of comparisons. However, the actual comparisons are not clear and there are no such symbols on the graphs themselves for this figure. Similarly, one of the symbols is used in figure 5, and these symbols are used in figure 4, without a symbol key. The figure legend for figure 6 lists statistically significant comparisons that are indicated on any of the graphs. Symbols should not be used in figure titles. This is just sloppy data presentation.

All corrected. Figures 4, 5 and 6 have been redone. We now use a different symbol (§) to indicate significance of the main fixed effect of Treatment. This was previously the same symbol (*) used for post-hoc tests, thus it was confusing. The reason for there being no such symbols on Figure 6 was that those individual tests within groups did not rise to significance, despite the main effects or interactions of main effects being significant. This is a normal occurrence in mixed-effects models where fixed and random effects absorb the variance and are significant for their distributed differences, yet with the control for multiple tests individual post-hoc tests don’t rise to significance. There was a key for the fixed effect and interaction symbols in panel 4B of the prior version of the figure. We placed it in panel 4B because the font would be too small as part of a combined legend in 4A. For further clarity, we now also define these symbols in the caption text of Figure 4. We have removed the used of these symbols in the titles of Figures 5 and 6. For Figure 5, we have removed the legend defining these symbols, and only have legends for symbols in the figure.

Issue 2: It is not appropriate to include data in a manuscript for which methods are not described or which have been previously published. Figure 7E-H should therefore be removed and a description of these results placed in the discussion  Unlike the results  for our previous hemisection injury model which showed ...

So, we had added the panels in Figure 7E-H in response to your previous criticism, which we agreed with, along with more discussion of the methods of the prior manuscript (in sections 1, 3, 4.1, and 4.2 of revised manuscript). We agreed with your point that it was confusing to refer to a prior study without showing the data:

2d) Titling a section and figure: ” 3.2. Comparison with hemi-section injury model (Fig. 4)” makes no sense. The authors do not show any comparison between data from this and the previous model and they only show a subset of data that they also show for the previous model. Furthermore, all discussions of comparison between data from this study and their previous study belongs in the Discussion.

So in order for the reader not to have to refer back to that paper, we pulled the original data set from that paper and re-plotted the relevant variables as panels 7E-7H in the revised figure. We think this, along with the additional explanation of the prior methods mentioned above, along with moving the comparison to the discussion, make the paper stronger, so we thank the reviewer for this suggestion.

When original source data are used to re-plot a figure, as was done here, we believe it is appropriate and journals encourage this, as long as the earlier paper is cited, as we have done here (with “Panel E-H data re-plotted from those gathered for Eisdorfer et al., 2022”) in the figure caption. As journals own the copyright to the figure, not the original data, it is not a problem if the original data are re-plotted, as long as the figure is not copied and used, which we did not do.

In sum, we much prefer this version that the reviewer suggested in their first review and would like to use it.

Line 94: I’m not sure what “scientific cutting” power is supposed to mean.

This refers to how much a hypothesis or experimental design has the ability to test (cut) between alternate hypotheses and/or answer scientific questions in a rigorous manner.

Line 130-131: missing comas “We utilized a similar paradigm, but with a hemi-section injury model, in our prior work [21], and thus a secondary goal of this work was to compare....”

Added.

Line 418: clarify wording....... increase in step height (no increase in depression of step height)

Corrected.

Line 462: I think that you mean to say that wave forms for DREADD and control animals differed

Clarified.

Line 469: “when analyzed at the rat rather than stride level” is confusing. I think that you mean when analyzed for individual animals rather than for all animals in each treatment groups.

Clarified the first time we discuss the rat versus stride level analysis above.

Line 474-475: This is confusing “Withdrawal of afferent excitation during kinematic recording in week seven did not introduce statistically significant changes in the data. Mean ankle height during swing showed a trend back towards the injured state (Fig.6ACE).”

Reworded.

Of course, adding new data would not change the data. Do you mean that it did not change measured parameters compared to the previous week? Animals are still in an injured state. Do you mean that mean ankle height in DREADD animals moved towards the values for the control treated group?

Clarifed per above – the last interpretation is correct. We meant that the ankle height in the DREADDs animals moved towards the values that both groups displayed in the earlier weeks of acute injury (week 2). We now state this.

Line 474-478: In section 3.4, Readers are required to go backwards to look at 2 previous figures. It would be better to make these points when the figures were first mentioned. It gets confusion to go back and revisit data that was previously analyzed.

We added relevant statements at the first mentions.

Line 483: missing coma : “...whereas at other speeds, the groups did not differ.” Line 502: As above, not as before.

Corrected.

Line 506: There is no figure 1I.

Corrected to Figure 4I.